# Sports Participation and Academic Performance in Primary School: A Cross-Sectional Study in Chinese Children

**DOI:** 10.3390/ijerph20043678

**Published:** 2023-02-19

**Authors:** Yao Zhang, Jin Yan, Xiao Jin, Hongying Yang, Ying Zhang, Huijun Ma, Rui Ma

**Affiliations:** 1Institute of Sports and Health, Zhengzhou Shengda University, Zhengzhou 451191, China; 2Centre for Active Living and Learning, University of Newcastle, Callaghan, NSW 2308, Australia; 3College of Human and Social Futures, University of Newcastle, Callaghan, NSW 2308, Australia; 4Department of Physical Education, Tianjin University of Technology, Tianjin 300384, China; 5Library of Beijing Sport University, Beijing 100084, China; 6School of Psychology, Shenzhen University, Shenzhen 518060, China; 7School of Physical Education, Taiyuan University of Technology, Taiyuan 030024, China; 8Postdoctoral Research Station in Public Administration, School of Physical Education, Zhengzhou University, Zhengzhou 450001, China

**Keywords:** sports participation, academic performance, children, primary school, China

## Abstract

Previous studies have demonstrated that the effect of sports participation on student health and academic performance is significant. However, the relationship between sports participation and academic performance in specific subjects (e.g., English) in the Chinese population is not clear, especially in primary schools. Therefore, the present cross-sectional study aimed to investigate the relationship between sports participation and academic performance in Chinese elementary schools. Methods: All study participants were asked to self-report their sociodemographic factors (e.g., sex, grade, age), independence, and outcomes. Alongside that, a self-reported questionnaire was used to assess participation in sports and academic performance of three core subjects in China’s schooling system (Chinese; math; English; from A to F, with A indicating the best academic performance). An ordered logistic regression, with an odds ratio (OR) at 95%CI confidence interval, was performed to examine the association between sports team participation and academic performance. Results: The final analysis included 27,954 children aged 10–14. Children in the fifth and sixth grades accounted for 50.2% and 49.8%. Chinese, math, and English academic performance were positively correlated with participation in sports. Compared with students who never participate in sports, those students who participate in sports 1–3 times a month, 1–2 times a week, and 3 or more times a week, were more likely to achieve better grades. In terms of math, compared with students who never participate in sports, those students who participate in sports 1–3 times a month, 1–2 times a week, and 3 or more times a week, were more likely to achieve better grades. Regarding English, compared with students who never participate in sports, those students who participate in sports 1–3 times a month, 1–2 times a week, and 3 or more times a week, were more likely to achieve better grades. Conclusions: Consistent with previous studies, the current study confirms the positive effect of sports participation on children’s academic performance. For an academic-related outreach, gender-, grade- and area-specific strategies should be considered in future research.

## 1. Introduction

Previous studies have demonstrated that the effect of sports participation on student health and academic performance is clear [1,2,3,4]. First, from a biological point of view, exercise produces dopamine, serotonin, and norepinephrine, three neurotransmitters associated with physical activity [5]. For example, high oxygen consumption during exercise increases blood circulation, increasing the number of muscle microvessels through an oxygen demand [6]. The faster the blood flows to the brain, the more the oxygen-carrying heme can be transported to the cells, allowing for more thinking and storage of more memories [7]. Proper physical activity can improve the balance and inhibition of neural processes, and improve the comprehensive ability of brain analysis [8]. At the same time, physical activities can also improve the nervous system’s regulating effect on internal organs, make organs and systems more flexible, and improve the body’s working ability based on the ability to resist diseases, so that students would not have too many health factors restricting their learning and growth [9,10]. Exercise is also related to emotion: exercise can inhibit the activation of the amygdala in the brain, and exercise is a good catharsis channel, which can let the brain secrete dopamine so that children have positive emotions [11,12]. Ishihara et al. (2020) [13] argue that some students are not interested in sports activities because of the heavy burden, which gives students little chance to relax and recreate. Similarly, Rees and Sabia (2021) [2] discuss the importance of exercise in correcting children’s mental perceptions. They believe that physical education is a kind of life attitude training, as physical fitness teaches students to cultivate an indomitable fighting spirit with the right will to deal with competition, so it is of great help to cultivate discipline, unity, and cooperation spirit [2].

Previous studies have suggested that sufficient physical activity (at least 60 min per day, or meeting the physical activity guidelines) [14,15,16], was linked to academic achievement among children [17]. The kind of physical activity is another qualitative characteristic that needs to be further explored [18]. Studies have shown that different exercise programs have different effects on physical fitness and have different benefits on children’s cognitive function. Chu and Zhang (2018) [19] discuss the more significant influence of aerobic exercise on the improvement of the following work. In particular, running can not only enhance the human metabolism and basal metabolic rate, but also improve the quality of sleep to ensure continuous learning [20]. Fedewa et al. (2011) [21] suggested that skipping rope could be used as an optimal choice for physical education in primary and secondary schools, which could not only ensure safety, but also exercise primary and secondary school students’ jumping ability and hand, eye, brain, and leg coordination. Many scholars do not recommend swimming as a physical examination item for teenagers, because swimming has certain safety problems. However, Hagiwara and Sport (2017) [22] suggest that it can be an important means to prevent drowning accidents and contribute more to the strengthening of the muscles and the whole body. Hastie et al. (2011) [23] found no significant difference in the reduction in all-cause mortality between aerobic exercise and strength training, nor in the adjustments to children’s cognitive function. Swing sports, such as baseball or badminton, stimulate the shoulder muscles and arms. In addition, swing sports also stimulate the brain to think quickly and nervously, which plays a role in strengthening and waking up the brain. However, for different ages and gender, the appropriate type of exercise, that enables students to adhere to the exercise long-term and has less wear on the body, is also different [24].

In previous studies on the relationship between sports participation and academic performance, Chen et al. (2021) [3] used a questionnaire to survey junior high school students and concluded that the frequency of weekly participation in sports was directly proportional to their academic performance, and the relationship was more obvious among girls. Similarly, Hut et al. (2021) [25] conducted a comparative experiment between groups that participated in sports and those that never participated in sports, and found that the groups that participated in sports generally showed sustained concentration, which led to a greater advantage in performance evaluation. Another study found a significant relationship between standardized verbal scores and physical activity, but a less strong relationship between math and physical activity [17,26]. Other studies have conservatively confirmed that there is no negative link between math performance and sports participation, and that interventions on students’ cognitive awareness and learning attitudes are needed to effectively improve academic performance [27]. In the analysis of the reasons for the proportional relation as the starting point, Chen et al. (2021) [3] thought that the brain white matter of the children who often take part in physical exercise, as compared to sedentary children who move less, had a more perfect network structure and a hippocampus that is more complex, allowing for connections of new neurons with existing connections between neurons more closely, and for more efficient brain activity during learning, memory and thinking. Post-exercise students were less likely to report anxiety and irritability when engaged in academic work, and were more focused on their studies after rest, thus showing less poor academic performance than sedentary, stressed students [28].

However, the relationship between sports participation and academic performance in the Chinese population is not clear, especially in primary schools. Therefore, the present cross-sectional study aimed at delving into the relationship between sports participation and academic performance in Chinese elementary schools.

## 2. Materials and Methods

### 2.1. Method

#### 2.1.1. Participants

A large-scale self-report survey was conducted in Shenzhen, in March 2021, China, by using convenience sampling. The target respondents were the students at local public primary schools. The survey was conducted in each district of Shenzhen. Grades 1–3 were not included in this study because of the students’ limited cognitive ability to take part in the study survey. Grade 6 students were also excluded because of their preparation for the middle school entrance examination. As a result, only grade 5 students were included in the study.

Before the survey was conducted, all the students and their parents were informed of the general information and purpose of the survey. It was emphasized that the students were free to decide whether to take the survey or not, and the data collected in this way would be kept anonymous during collection and analysis. The students completed the questionnaires online, using computers in the schools independently. The questionnaires were handed out to the students through an online platform of the questionnaire (https://www.wjx.cn/, accessed on 1 December 2022). Before going to the page of formal response, the electronic page of informed consent was presented to the respondents. This study was conducted based on an approval from the Research Committee of Shenzhen University (No. 2020005) and schools that participated in the survey. The approval date was 21 May 2020.

#### 2.1.2. Measurements

(1) Main variables

The sports participation frequency was measured by the setting question “how often did you participate in sports teams or sports clubs in the last 12 months?” There were several possible options for students, that were: never, three times or more a month, once or twice a week, and once or three times a week. Detailed information on the questionnaire, scoring, validity, and reliability of the study measures can be found elsewhere [3]. Sports participation is based on the item of the Youth Risk Behavior Survey (YRBS); this item has been fully applied in different countries and has sufficient reliability and validity. More information regarding YRBS procedures can be found by accessing the Centers for Disease Control and Prevention website.

The students also reported their academic performances in English, math, and Chinese in the last exam. in China, there are 6 grades to evaluate the student’s performance in the examinations they take. In detail, the best examination performance is rated A, while the worst performance was rated F. Similar questions and options could be found in the Youth Risk Behavior Survey, which is designed to check the academic performance of students, and was widely adopted in previous research [29,30].

(2) Covariates

The data on the following information was also collected: sex (boy or girl), grade (5th or 6th), age (10 or 11 or 12 or 13 or 14), subjective family socioeconomic status (a proxy measure to assess study participants’ socioeconomic status, which was measured using the adapted version of the MacArthur Scale of Subjective Social Status [31]) (low or middle or high), parents’ educational background (master or above; bachelor or equivalent; high school or equivalent; junior middle school or below; unknown), siblings (only child/have brothers or sisters). These factors were treated as covariates in further statistical analysis.

#### 2.1.3. Data Analyses

Data from individual questionnaires were filtered for missing or implausible values. A total of 27,954 respondents provided complete data on the variable of interest needed in the study, and they had been included in the study for final analysis. STATA BE 17.0 (College Station, TX, USA) was adopted to perform the statistical analysis. Missing data have been addressed by complete cases before formal analysis. In the statistical analysis, only the age variable was treated as a continuous variable, while the others were treated as nominal or categorical variables. Descriptive statistics were made to describe the sample characteristics by presenting the mean (m) with standard deviation (sd) for continuous variables, or frequency (n) with percentage (%) for categorical variables. In the statistical analysis, SES was categorized into three levels: low, medium and high. AA was reclassified into A, B, C and D (merged E and F into D). Finally, an ordered logistic regression, with an odds ratio (OR) at 95%CI confidence interval, was performed to examine the association between sports team participation and academic performance after controlling for the covariates, including sex (reference group: boy), age (reference group: being youngest), grade (reference group: grade 5), only one child or not (reference group: not), father or mother education levels (reference group: unknown) and SES level (reference group: low). Three models were fitted into examining the associations of sports participation of Chinese, math and English. For sports participation, the level of never (the lowest) was treated as the reference group. For achievements in Chinese, math, and English, the level of D (the poorest) was treated as the reference group. Further, similar sets of regression models by sex were also established for Chinese, math, and English, separately. Results were expressed as an odds ratios (OR), and the confidence interval (CI) was 95%. A prior *p*-value below 0.05 has been adopted to determine statistical significance. 

## 3. Results

The demographic characteristics of the total sample in the current study were presented in Table 1. A total of 27,954 children were involved in the final analysis. The mean age of all the study participants was 11.5 (±0.8), with boys accounting for 53.3%. The children of the two grades were 50.2% (fifth graders) and 49.8% (sixth graders), respectively. As for the parental highest education level, most of the population was “undergraduate”, accounting for 37.8% for fathers and 35.9% for mothers. Around 80% of the study sample reported grades of A or B. Further information can be found in Table 1.

The demographic characteristics of the sports participation frequency were displayed in Table 2. It can be seen that children surveyed frequently to “Never” participate in sports made up most of the population among both boys and girls, also accounted for a similar percentage between grade 5 (47.9%) and grade 6 (52.1%). In terms of age, 11- and 12-years-olds account for the largest number of boys and girls. More detailed information can be observed in Table 2.

Overall, the frequency of participation in sports was positively associated with a better academic performance in Chinese, math, and English. According to Table 3, compared with students who never participate in sports, those students who participate in sports 1–3 times a month, 1–2 times a week, and 3 or more times a week were more likely to achieve better grades in Chinese learning performance (OR = 1.14, 95% CI: 1.07–1.21; OR = 1.17, 95% CI: 1.10–1.26; and OR = 1.31, 95% CI: 1.22–1.40, respectively). A similar situation was also shown for boys: compared with boys who never participate in sports, those who participate in sports 1–3 times a month, 1–2 times a week, and 3 or more times a week were more likely to achieve better grades Chinese learning performance (OR = 1.12, 95% CI: 1.03–1.22; OR = 1.19, 95% CI: 1.10–1.30; and OR = 1.44, 95% CI: 1.32–1.57, respectively). In terms of girls, compared with students who never participate in sports, those students who participate in sports 1–3 times a month, 1–2 times a week, and 3 or more times a week were more likely to achieve better grades in Chinese learning performance (OR = 1.16, 95% CI: 1.06–1.26; OR = 1.15, 95% CI: 1.05–1.27; and OR = 1.13, 95% CI: 1.02–1.26, respectively).

The associations between sport participation and Math learning performance were shown in Table 4. Compared with students who never participate in sports, those students who participate sports in 1–3 times a month, 1–2 times a week, and 3 or more times a week were more likely to achieve better grades in Math learning performance (OR = 1.15, 95% CI: 1.09–1.22), (OR = 1.16, 95% CI: 1.09–1.24) and (OR = 1.28, 95% CI: 1.20–1.37), namely. In terms of boys, compared with boys who never participate in sports, those boys who participate sports in 1–3 times a month, 1–2 times a week, and 3 times a week and above were more likely to achieve better grades in Math learning performance (OR = 1.14, 95% CI: 1.05–1.24), (OR = 1.24, 95% CI: 1.14–1.35) and (OR = 1.38, 95% CI: 1.26–1.52). Regarding girls, compared with students who never participate in sports, those students who participate sports in 1–3 times a month, 1–2 times a week, and 3 times a week and above were more likely to achieve better grades in Math learning performance (OR = 1.17, 95% CI: 1.07–1.27), (OR = 1.08, 95% CI: 0.99–1.19) and (OR = 1.17, 95% CI: 1.06–1.30).

Table 5 presented the associations between sport participation and English learning performance. Compared with students who never participate in sports, those students who participate sports in 1–3 times a month, 1–2 times a week, and 3 times a week and above were more likely to achieve better grades in English learning performance (OR = 1.14, 95% CI: 1.08–1.21), (OR = 1.25, 95% CI: 1.18–1.33) and (OR = 1.35, 95% CI: 1.26–1.44), namely. In terms of boys, compared with boys who never participate in sports, those boys who participate sports in 1–3 times a month, 1–2 times a week, and 3 times a week and above were more likely to achieve better grades in English learning performance (OR = 1.18, 95% CI: 1.09–1.28), (OR = 1.38, 95% CI: 1.27–1.50) and (OR = 1.52, 95% CI: 1.39–1.66). Regarding girls, compared with students who never participate in sports, those students who participate sports in 1–3 times a month, 1–2 times a week, and 3 times a week and above were more likely to achieve better grades in English learning performance (OR = 1.10, 95% CI: 1.01–1.20), (OR = 1.11, 95% CI: 1.01–1.22) and (OR = 1.14, 95% CI: 1.02–1.26).

## 4. Discussion

Based on previous studies, the purposes of the present research include the examination of the relationship between academic performance and sports participation, providing more academic evidence, and confirming the positive effect of sports team participation on cognitive outcomes among primary school students in China. In summary, the present study found that sports participation could improve the academic performance of children.

The importance of sport participation for math is often reflected in the cognitive functions of the brain [32], with the production of dopamine, serotonin, and norepinephrine during exercise, three neurotransmitters associated with the cerebellum’s ability to decode and encode [33]. As early as 2013, a study found that students with weak athletic skills also performed poorly in numeracy, and that performing body movements while studying math improved math scores in the lower grades of elementary school [34]. For younger students, in particular, those who scored lower on tests of flexibility, speed, and sensitivity of hands scored lower on tests of reading and arithmetic. From a neuroscientific point of view, moderate to vigorous physical activity is associated with white matter health, and as brain proteins improve, so do neural connections and conduction in different brain regions [35]. Daily aerobic exercise can also increase the volume of cerebral grey and can improve cognitive function and creativity. Because exercise optimizes the structural and functional connections of the brain hemispheres, it can improve both motor skills and mathematical thought processes. This is confirmed by Dumais’s (2008) [36] “zero-hour sports program,” in which students arrive at school at 7 a.m. and exercise before classes begin, until their heart rate is at its peak or 70% of their VO2 max. The experiment found that the experimental group who took math courses in the morning improved their learning more than twice as much as the control group who took math courses in the afternoon [36]. This is because neurotransmitters produced after exercise play an important role in learning.

In addition, according to previous studies, sports play a more significant role in language and other liberal arts learning, which is due to the positive effect of sports on attention and memory [37]. After exercise, a large amount of serotonin is produced, which is related to memory and emotion, so that the learning effect can be greatly improved [38]. At the same time, the secretion of norepinephrine and dopamine has a direct relationship with attention, so students’ concentration is enhanced. Therefore, with proper exercise, students would not only become happy, but also pay more attention in class. After receiving positive feedback from their study, their grades would continue to improve. In a controlled study, students who participated in regular exercise had about 10% better reading and comprehension skills than those who only took regular gym classes [39]. Therefore, past studies emphasize that exercise promotes new brain cells and connections between nerve cells in the brain, enhancing brain plasticity, which is conducive to the development of language sense. For example, the brain’s processing of letters is controlled by two different parts [40]. The right brain is used to depict images, while the left brain symbolizes the observed content. Therefore, the flexible switching of brain functions is necessary for language teaching. Alongside that, exercise can enhance heart and lung function, and a larger lung capacity can make students pronounce words more smoothly [41]. This indicates that the good development of articulation organs can improve articulation clarity to some extent and cultivate confidence in language learning at a young age as much as possible. At the same time, inactive children generally have motor development delays, their social and emotional ability would also be delayed, and their frequency of communication with others is also a major constraint on language development [42]. Therefore, a larger range of activities means an increase in social coverage. Another interesting finding is that girls did not find a significant dose relationship in Chinese and math, which may be caused by gender differences and subject differences. With an increase in sports participation, the dose response of different genders to academic performance is different [43]. The same variables, therefore, lead to this difference. Thus, there were differences in the dose relationships across subjects between boys and girls. In future research, more high-quality research should be devoted to improving girls’ self-perceptions and well-being during sports participation.

Taking China as the research background, it was found that schools face great challenges in allocating physical exercise time. According to the 2007 annual report, 53.6% of American high school students participate in sports for 1 day or more per week on average, while only 17.1% of Chinese high school students meet the current physical exercise recommendations [44]. Foo and other scholars (2009) [45] suggest that the main way to improve the frequency of sports participation of Chinese students is to include sports performance in the quality assessment, and the mandatory admission conditions can guarantee the basic sports participation of students. On the contrary, Peter believes that such performance-oriented sports activities will continue to increase the pressure on Chinese students and have great limitations in terms of physical function. Therefore, some scholars believe that the policy reform of the educational structure can interfere with the sports tendency of teenagers [46,47]. For example, the diversification of school sports programs can stimulate the enthusiasm of students to participate in sports, so that students can make spontaneous and personalized choices. In addition, Wang and Dou (2020) [47] have discussed the importance of families in conveying athletic ability. Parents who maintain long-term athletic goals are more likely to have children who develop healthy sports habits and can provide a deeper understanding of the relationship between sports and academic performance in adolescents.

The relationship between physical exercise and adolescent cognitive ability starts from the arousal theory in environmental psychology, which idles on an inverted U-shaped relationship between individual performance and arousal level [48]. Thus, Castelli and other scholars (2014) [49] analyzed that higher levels of physical activity are better for optimal levels of arousal to be reached, but that cognitive and executive performance begins to decline gradually after peak physical strength is reached. Previous research indicates that participation in intermittent high-intensity activities can have long-term benefits on cognitive function and also proves that a link exists between high-intensity exercise and cognitive performance in adolescents on the basis of the retrospective study [50]. For example, Leahy’s (2020) [51] study has shown that participation in high-intensity interval training can improve the psychological quality of children, thereby improving the cognitive function and mental health of adolescents, and that the acute effect of HIIT is generally stronger than the cognitive function caused by chronic effects. Furthermore, Pesce [52] proposed that qualitative features of exercise may have different influences on cognition (i.e., coordination complexity and cognitive need). In other words, exercises requiring the allocation of more attention and cognitions (i.e., cognitive engagement) are thought to improve cognitive performance more than those with less cognitive engagement (e.g., continuous running) [53]. The above-mentioned “cognitive stimulation viewpoint” states that activities requiring high coordination will stimulate similar brain areas to those responsible for higher-order cognitive processes [54]. Another study by Chang et al. (2011) [55] took 42 teenagers as samples, and defined the treatment group as 30 min of medium-high intensity bicycle riding and the control group as 30 min of reading training. The comparative experiment provided a basis for short-term aerobic training to improve teenagers’ executive ability. Chaddock-Heyman et al. (2013) [56] found that adolescents aged 8–9 years, who exercised 5 times a week for at least 1 h, showed improved performance in attention and anti-interference ability after continuous exercise for 9 months.

Dyer and other scholars used longitudinal research methods to find a positive relationship between sports participation and academic achievement [32]. Still, the results diverged when gender or parental education was taken into account. A previous study [57] analyzed that exercise participation improved memory, showing a positive correlation with academic performance [32]. Similarly, Burns and other scholars [29] suggest that students who are able to build exercise participation into their lives have higher health indicators, such as sleep levels and whether they eat breakfast, and that the combination of factors is key to academic performance. Using the control group experiment, Howie and Pate (2012) [58] proposed that although the positive model between sports participation and academic performance could not be drawn, the negative impact of non-sports participation on academic performance could be confirmed, especially for mathematics: non-sports participation students could significantly improve their math and logic ability. Due to the small sample sizes of the above-mentioned literature, they are affected by different backgrounds, so the results are not representative. Academic performance is also influenced by factors such as family and gender, as well as other health habits, which cannot be fully taken into account, and thus influence the determination of the relationship [59]. Future studies need to set different experiments according to different variables to ensure the accuracy of the results.

However, it is necessary to acknowledge the limitations on self-reporting of academic performance, because it is prone to recall bias and social desirability. Secondly, it is likely that the causal relationship could not be explained satisfactorily by the cross-sectional design. Therefore, the experimental and prospective research design should be adopted to gain a better understanding of the causal relationship between academic performance and sports participation in elementary schools, to enhance the intervention success rate. Thirdly, some bias might exist due to the use of a questionnaire for data collection. Finally, the obvious disadvantage of convenience sampling is that it is likely to be biased, as it may not be representative of the population. Considering the limitations, future studies should be made to solve the problem and gain stronger evidence.

The findings of this study have some practical implications for future studies. Firstly, the government, schools, and families need to pay attention to the positive feedback of sports participation on the cognitive development and academic performance of adolescents, and the efforts of the three parties are needed to intervene. The government should solve the defects of unbalanced sports facilities and unreasonable education structures to provide perfect hardware conditions for teenagers. Secondly, schools need to make reasonable sports plans and breaks to properly mobilize students’ focus and enthusiasm, so that students can feel the positive feedback of sports participation as soon as possible. For instance, autonomy may be satisfied by offering participants selections of the exercises or work and rest intervals. Apart from that, providing positive feedback and encouragement can be helpful to strengthen the competence awareness of participants. Conducting activities with friends tends to be more enjoyable and meets the relatedness perception. The cognitive health benefits can be optimized through other strategies, including taking advantage of music and exercising in a natural environment, etc. Finally, in the family, parents need to actively guide and timely capture the interests of teenagers, to maximize the children’s physical needs. Therefore, sports education for young people is a long-term strategic goal, which not only needs a high-quality macro condition, but also needs to take into account the individual needs of students, to truly promote the participation of students in sports and play a positive role in academic performance.

## 5. Conclusions

This study has offered evidence for the positive relationship between sports participation on children’s academic performance in Chinese primary schools. Further study should be given attention to the government, schools, and families, to offer positive feedback of sports participation on the cognitive development and academic performance of adolescents. Moreover, schools need to make reasonable sports plans and breaks to properly mobilize students’ focus and enthusiasm, so that students can feel the positive feedback of sports participation as soon as possible. Finally, it is recommended that school-aged children are allocated to participate in cognitively challenging physical activities, such as tag games, team games, or coordinative exercises, which positively affect executive functions and improve academic performance in a school-based setting.

## Figures and Tables

**Table 1 ijerph-20-03678-t001:** Demographic characteristics of study participants.

		Total	
		n	%
Total			100.0
Sex		27,954	
	Boy	14,892	53.3
	Girl	13,062	46.7
Grade			
	5th	14,026	50.2
	6th	13,928	49.8
Academic performance (Chinese)			
	A	12,968	46.4
	B	10,753	38.5
	C	3479	12.4
	D	754	2.7
Academic performance (Math)			
	A	13,502	48.3
	B	8577	30.7
	C	4162	14.9
	D	1713	6.1
Academic performance (English)			
	A	12,683	45.4
	B	8854	31.7
	C	4610	16.5
	D	1807	6.5
Siblings			
	Yes	6518	23.3
	No	21,436	76.7
Father Education Level			
	Middle school or below	6131	21.9
	High school	6948	24.9
	Undergraduate	10,557	37.8
	Graduate	1232	4.4
	Unknown	3086	11.0
Mother Education Level			
	Middle school or below	7143	25.6
	High school	7070	25.3
	Undergraduate	10,039	35.9
	Graduate	790	2.8
	Unknown	2912	10.4
Socioeconomic status			
	Low	5386	19.3
	Middle	20,708	74.1
	High	1860	6.7
Sports participation			
	Never	12,233	43.8
	1–3 times a month	6112	21.9
	1–2 times a week	5347	19.1
	3 or more times a week	4262	15.2
		Means	SD
Age		11.5	0.8

**Table 2 ijerph-20-03678-t002:** Demographic characteristics of the participants according to sports participation frequencies and sex.

Demographic Characteristics		Boys				Girls			
		Never	1–3 Times a Month	1–2 Times a Week	3 or More Times a Week	Never	1–3 Times a Month	1–2 Times a Week	3 or More Times a Week
Grade	Grade5	3056 (47.9%)	1585 (52.2%)	1509 (51.4%)	1276 (50.2%)	2868 (49.0%)	1573 (51.1%)	1255 (52.1%)	904 (52.5%)
	Grade6	3322 (52.1%)	1451 (47.8%)	1428 (48.6%)	1265 (49.8%)	2987 (51.0%)	1503 (48.9%)	1155 (47.9%)	817 (47.5%)
Age	10	509 (8.0%)	242 (8.0%)	268 (9.1%)	202 (7.9%)	523 (8.9%)	298 (9.7%)	221 (9.2%)	160 (9.3%)
	11	2402 (37.1%)	1202 (39.6%)	1177 (40.1%)	939 (37.0%)	2309 (39.4%)	1203 (39.1%)	987 (41.0%)	666 (38.7%)
	12	2800 (45.2%)	1346 (44.3%)	1283 (43.7%)	1164 (45.8%)	2501 (42.7%)	1313 (42.7%)	984 (40.8%)	731 (42.5%)
	13	551 (8.6%)	234 (7.7%)	196 (6.7%)	221 (8.7%)	481 (8.2%)	247 (8.0%)	205 (8.5%)	154 (8.9%)
	14	36 (0.5%)	12 (0.3%)	13 (0.5%)	15 (0.6%)	41 (0.7%)	15 (0.5%)	13 (0.5%)	10 (0.6%)
Academic Performance (Chinese)	A	2347 (36.8%)	1227 (40.4%)	1276 (43.4%)	1216 (47.9%)	2919 (49.9%)	1704 (55.4%)	1343 (55.7%)	936 (54.4%)
	B	2643 (41.4%)	1276 (42.0%)	1190 (40.5%)	949 (37.3%)	2192 (37.4%)	1068 (34.7%)	838 (34.8%)	597 (34.7%)
	C	1117 (17.5%)	422 (13.9%)	387 (13.2%)	320 (12.6%)	633 (10.8%)	252 (8.2%)	192 (8.0%)	156 (9.1%)
	D	271 (4.2%)	111 (3.7%)	84 (2.9%)	56 (2.2%)	111 (1.9%)	52 (1.7%)	37 (1.5%)	32 (1.9%)
Academic Performance (Math)	A	3011 (47.2%)	1562 (51.4%)	1618 (55.1%)	1442 (56.7%)	2504 (42.8%)	1457 (47.4%)	1112 (46.1%)	796 (46.3%)
	B	1944 (30.5%)	904 (29.8%)	850 (28.9%)	717 (28.2%)	1819 (31.1%)	993 (32.3%)	792 (32.9%)	558 (32.4%)
	C	983 (15.4%)	422 (13.9%)	340 (11.6%)	284 (11.2%)	1061 (18.1%)	445 (14.5%)	364 (15.1%)	263 (15.3%)
	D	440 (6.9%)	148 (4.9%)	129 (4.4%)	98 (3.9%)	471 (8.0%)	181 (5.9%)	142 (5.9%)	104 (6.0%)
Academic Performance (English)	A	2350 (36.8%)	1275 (42.0%)	1413 (48.1%)	1261 (49.6%)	2716 (46.4%)	1554 (50.5%)	1253 (52%)	861 (50.0%)
	B	2039 (32.0%)	1001 (33.0%)	884 (30.1%)	757 (29.8%)	1884 (32.2%)	986 (32.1%)	739 (30.7%)	564 (32.8%)
	C	1304 (20.4%)	550 (18.1%)	483 (16.4%)	388 (15.3%)	919 (15.7%)	414 (13.5%)	328 (13.6%)	224 (13.0%)
	D	685 (10.7%)	210 (6.9%)	157 (5.3%)	135 (5.3%)	336 (5.7%)	122 (4.0%)	90 (3.7%)	72 (4.2%)
Siblings	Yes	1568 (24.6%)	793 (26.1%)	836 (28.5%)	625 (24.6%)	1182 (20.2%)	653 (21.2%)	538 (22.3%)	323 (18.8%)
	No	4801 (75.4%)	2243 (73.9%)	2101 (71.5%)	1916 (75.4%)	4673 (79.8%)	2423 (78.8%)	1872 (77.7%)	1398 (81.2%)
Father Education Level	Middle school or below	1632 (25.6%)	648 (21.3%)	577 (19.6%)	479 (18.9%)	1386 (23.7%)	578 (18.8%)	455 (18.9%)	376 (21.8%)
	High school	1609 (25.2%)	807 (26.6%)	710 (24.2%)	639 (25.1%)	1381 (23.6%)	812 (26.4%)	601 (24.9%)	389 (22.6%)
	Undergraduate	2047 (32.1%)	1169 (38.5%)	1270 (43.2%)	1007 (39.6%)	2059 (35.2%)	1273 (41.4%)	1049 (43.5%)	683 (39.7%)
	Graduate	195 (3.1%)	136 (4.5%)	177 (6.0%)	155 (6.1%)	218 (3.7%)	137 (4.5%)	128 (5.3%)	86 (5.0%)
	Unknown	895 (14.0%)	276 (9.1%)	203 (6.9%)	261 (10.3%)	811 (13.9%)	276 (9.0%)	177 (7.3%)	187 (10.9%)
Mother Education Level	Middle school or below	1894 (29.7%)	733 (24.1%)	630 (21.5%)	558 (22.0%)	1689 (28.8%)	689 (22.4%)	523 (21.7%)	427 (24.8%)
	High school	1591 (24.9%)	825 (27.2%)	778 (26.5%)	643 (25.3%)	1325 (22.6%)	849 (27.6%)	628 (26.1%)	431 (25.0%)
	Undergraduate	1927 (30.2%)	1116 (36.8%)	1231 (41.9%)	965 (42.1%)	1963 (33.5%)	1193 (38.8%)	1010 (41.9%)	634 (36.8%)
	Graduate	117 (1.8%)	92 (3.0%)	95 (3.2%)	124 (4.9%)	139 (2.4%)	85 (2.8%)	78 (3.2%)	60 (3.5%)
	Unknown	849 (13.3%)	203 (6.9%)	203 (6.9%)	251 (9.9%)	739 (12.6%)	260 (8.5%)	171 (7.1%)	169 (9.8%)
Socioeconomic Status	Low	1447 (22.7%)	581 (19.1%)	474 (16.1%)	389 (15.3%)	1298 (22.2%)	504 (16.4%)	396 (16.4%)	297 (17.2%)
	Middle	4578 (71.8%)	2235 (73.6%)	2231 (76.0%)	1871 (73.6%)	4271 (72.9%)	2381 (77.4%)	1861 (77.3%)	1280 (74.4%)
	High	353 (5.5%)	220 (7.2%)	232 (7.9%)	281 (11.1%)	286 (4.9%)	191 (6.2%)	153 (6.3%)	144 (8.4%)

**Table 3 ijerph-20-03678-t003:** Odds ratio and 95% CI for academic performance in relation to sports participation (Chinese).

Overall					Boys					Girls				
Chinese	OR	95% CI		Sig.	Chinese	OR	95% CI		Sig.	Chinese	OR	95% CI		Sig.
Frequencies														
1–3 times a month	1.14	1.07	1.21	** *p* ** **< 0.001**	1–3 times a month	1.12	1.03	1.22	**0.004**	1–3 times a month	1.16	1.06	1.26	**0.001**
1–2 times a week	1.17	1.10	1.25	** *p* ** **< 0.001**	1–2 times a week	1.19	1.10	1.30	** *p* ** **< 0.001**	1–2 times a week	1.15	1.05	1.27	**0.002**
3 or more times a week	1.31	1.22	1.40	** *p* ** **< 0.001**	3 or more times a week	1.44	1.32	1.57	** *p* ** **< 0.001**	3 or more times a week	1.13	1.02	1.26	**0.017**

Reference group: Never participates in sports. All models controlled for sex, age, grade, subjective family socioeconomic status, parents’ educational background and siblings; statistically significant at *p* < 0.05.

**Table 4 ijerph-20-03678-t004:** Odds ratio and 95% CI for academic performance concerning the sports participation (Math).

Overall					Boys					Girls				
Math	OR	95% CI		Sig	Math	OR	95% CI		Sig	Math	OR	95% CI		Sig
Frequencies														
1–3 times a month	1.15	1.09	1.22	** *p* ** **< 0.001**	1–3 times a month	1.14	1.05	1.24	**0.002**	1–3 times a month	1.17	1.07	1.27	** *p* ** **< 0.001**
1–2 times a week	1.16	1.09	1.24	** *p* ** **< 0.001**	1–2 times a week	1.24	1.14	1.35	** *p* ** **< 0.001**	1–2 times a week	1.08	0.99	1.19	0.069
3 times a week and above	1.28	1.20	1.37	** *p* ** **< 0.001**	3 times a week and above	1.38	1.26	1.52	** *p* ** **< 0.001**	3 times a week and above	1.17	1.06	1.30	**0.002**

Reference group: Never participate in sports. All models controlled for sex, age, grade, subjective family socioeconomic status, parents’ educational background and siblings; statistically significant at *p* < 0.05.

**Table 5 ijerph-20-03678-t005:** Odds ratio and 95% CI for academic performance concerning the sports participation (English).

Overall					Boys					Girls				
English	OR	95% CI		Sig	English	OR	95% CI		Sig	English	OR	95% CI		Sig
Frequencies														
1–3 times a month	1.14	1.08	1.21	** *p* ** **< 0.001**	1–3 times a month	1.18	1.09	1.28	** *p* ** **< 0.001**	1–3 times a month	1.10	1.01	1.20	**0.025**
1–2 times a week	1.25	1.18	1.33	** *p* ** **< 0.001**	1–2 times a week	1.38	1.27	1.50	** *p* ** **< 0.001**	1–2 times a week	1.11	1.01	1.22	**0.018**
3 times a week and above	1.35	1.26	1.44	** *p* ** **< 0.001**	3 times a week and above	1.52	1.39	1.66	** *p* ** **< 0.001**	3 times a week and above	1.14	1.02	1.26	**0.013**

Reference group: Never participates in sports. All models controlled for sex, age, grade, subjective family socioeconomic status, parents’ educational background and siblings; statistically significant at *p* < 0.05.

## Data Availability

The raw data supporting the conclusions of this article will be made available by the authors, without undue reservation.

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
