# Peer review of "Sports Participation and Academic Performance in Primary School: A Cross-Sectional Study in Chinese Children"

_ijerph, 2023, doi:10.3390/ijerph20043678_

Round 1

Reviewer 1 Report

Dear researchers, first of all, congratulate you on this research, it is an interesting and relevant topic in school contexts. After reading this document, I had the following doubts about the paper:

1. Introduction:

- The research seeks to associate sports participation and academic performance; however, we know that there are different types of sports, also individual or collective sports, among other classifications that can be found. But I don't see this consideration that each sport has. On the other hand, during the introduction more benefits of exercise are mentioned, even of physical education class, but what happens with other sports? Sport contributes to physical activity, but also to physical exercise during its practice? regular?

- Sports participation is also multifactorial, as you mention, and we can find it as "barriers to sports practice", or physical-sports activities, but the benefits of exercise are developed in a large part of the research. And it is understood that sport is what is being studied along with academic performance.

- On the other hand, it is mentioned at the beginning of the instruction "many studies have demonstrated" and only 3 articles are cited. Please reconsider this statement. I found a similarity with this article

2.- Materials and methods:

- It is not clear to me, were only 5th grade students considered because they read and understand compared to 6th grade students?

- The page available, (https://www.wjx.cn/) is where you created the questionnaire?

- This instrument was adapted, but was this adaptation validated?

- This questionnaire has items, total number of questions, what is your name?

- What type of sports did you refer to in this questionnaire? Did you ask for practice time in minutes or hours?

- Was the sample selection for convenience? Was that also considered in the conclusions or limitations?

-This is the study of the ref. 3 for the methodology?, https://www.frontiersin.org/articles/10.3389/fpubh.2021.730497/full , I understand that this questionnaire was adapted?

3.- Results:

- Table 1 is not seen in its entirety.

- I suggest highlighting significant values ​​in bold.

4- Discussion:

- There is scientific literature, which indicates that sports practice improves cognitive performance, as well as executive function (in China and in other countries), this could support its discussion, but the benefits of  exercise are indicated.

- Reconsider the limitations and strengths.

5.- Conclusion: I think it is important to mention the type of sport that could contribute to improving academic performance in schoolchildren.

Others:

Review the citations and references, there are citations in the research that include the years and in another the year does not appear.

Ref. 7 is an investigation in adults.

Author Response

Responses to Reviewer 1 Comments

Dear reviewers,

Thank you for your time and valuable comments. We have provided a point-by-point response to each of your comments and suggestions and have made the appropriate changes to the manuscript. We believe the paper has improved significantly because of the review process.

Response to Reviewer comments

  1. Introduction:

- The research seeks to associate sports participation and academic performance; however, we know that there are different types of sports, also individual or collective sports, among other classifications that can be found. But I don't see this consideration that each sport has. On the other hand, during the introduction more benefits of exercise are mentioned, even of physical education class, but what happens with other sports? Sport contributes to physical activity, but also to physical exercise during its practice? regular?

Response: Thank you for the suggestion, please see the line 80-104

- On the other hand, it is mentioned at the beginning of the instruction "many studies have demonstrated" and only 3 articles are cited. Please reconsider this statement. I found a similarity with this article

Response: Thank you for the suggestion, we have fixed and added the new reference as requested. Please see the line 54-55.

2.- Materials and methods:

- It is not clear to me, were only 5th-grade students considered because they read and understand compared to 6th-grade students.

Response: Thank you for the suggestion. Please see the line 141-144.

- This instrument was adapted, but was this adaptation validated?

Response: 

Sports participation is based on the item of Youth Risk Behavior Survey (YRBS); this item has been fully applied in different countries and has sufficient reliability and validity. More information regarding YRBS procedures can be found by accessing the Centers for Disease Control and Prevention website.

- This questionnaire has items, total number of questions, what is your name?

Response: Thank you for the suggestion, please see the line 159-161.

- What type of sports did you refer to in this questionnaire? Did you ask for practice time in minutes or hours?

Response: Thank you for the suggestion, however, we have added the “Frequency of sport participation”, please see the line 160-163.

- Was the sample selection for convenience? Was that also considered in the conclusions or limitations?

Response: Thank you for the suggestion, we have added the limitation, please see the line 444-446.

-This is the study of the ref. 3 for the methodology?, https://www.frontiersin.org/articles/10.3389/fpubh.2021.730497/full , I understand that this questionnaire was adapted?

Response: Yes, our study and reference 3 have employed the same questionnaire for Sports participation.

3.- Results:

- Table 1 is not seen in its entirety.

Response: Thank you for the suggestion. Please see the table 1.

- I suggest highlighting significant values in bold.

Response: Thank you for the suggestion. Please see the table 2-4.

4- Discussion:

- There is scientific literature, which indicates that sports practice improves cognitive performance, as well as executive function (in China and in other countries), this could support its discussion, but the benefits of exercise are indicated.

Response: Thank you for the suggestion. Please see the line 384-414.

- Reconsider the limitations and strengths.

Response: Thank you for the suggestion, we have added the limitations in study, please see the line 436-439.

5.- Conclusion: I think it is important to mention the type of sport that could contribute to improving academic performance in schoolchildren.

Response: Thank you for the suggestion, we have added the type of sport in conclusion and abstract. Please see the line 478-482.

Others:

Review the citations and references, there are citations in the research that include the years and in another, the year does not appear.

Response: Thank you for the suggestion, we have updated the references.

Ref. 7 is an investigation in adults.

Response: Thank you for the suggestion, we have updated the reference.

Reviewer 2 Report

Summary of the article

The submitted paper titled, “Sport participation and academic performance in primary school: a cross-sectional study in Chinese children,” is cross-sectional study conducted in a large sample of Chinese school adolescents. The study is assessing he association between activity/sports participation and academics across different school subjects. The authors take care to detail the biologic plausibility of the tested associations and to explain the results presented. The manuscript is well written and may warrant publication after reviewer comments are addressed either by explanation or in changes to the manuscript.

Introduction/Background

11.       I would recommend that the language be changed in the following statement “Boring theoretical knowledge combined with psychological pressure will further affect their academic performance.” It sounds like the authors are stating that theoretical knowledge is boring. This can be offensive to academics and teachers who over theoretical concepts. It may be better to restate that students may perceive subjects as boring or change the word all together.

Methods

22.       The sports participation frequency variable operationalization was well explained; however, the authors state that the validity can be reviewed “elsewhere.” I would suggest that the authors provide some report of the prior validity given this is the primary independent variable of interest in all of the models.

33.       Can the authors clarify the academic performance variable. This is only the grade on a single exam or is this their grade in the class. Also please clarify if this is a self-reported grade. If this is self-report please indicate in the limitations and potential bias that can be inherent in self-reported exam grades.

44.       Regarding the modeling approach what are the authors reporting odds ratios for the linear models? Would it be simpler to report betas for the linear change? Please explain.

55.       Given that the data was collected in schools and classrooms did the authors consider multilevel models? There are two elements of concern: 1) the models may be identifying class differences or school differences that are unaccounted for. The clustering at the school or classroom level may be biasing the model effects and 2) given the large sample size are the models overpowered? Sometimes overpower in a model can drive significance. Was a power analysis run? Partitioning the groups in a multilevel model can also help here too. I am not saying this is a requirement just needs to be explained and maybe considered in the limitations if not addressed. What was the ICCs across schools or classrooms? Where they at an acceptable level.

Results

16.       Given that some of the Cis are not overlapping could some of these results suggest a dose-response relationship? Maybe this can be considered in the discussion. If so how much is a good amount?

Discussion

27.       I recommend that the authors take care in conclusive statements regarding the results given the cross-sectional nature of the study design. Example: “In sum, it is found in the present study that sports participation can effectively improve the academic performance of children.”

38.       Could the authors discuss some about the type of activity? I know that this was not the focus of the current studies results but, discussion about the types of activity or specifics about recommended activity durations per week/day.

49.       Consider adding some to the limitations based on prior comments from above.

Author Response

Responses to Reviewer 2 Comments

Introduction/Background

  1. I would recommend that the language be changed in the following statement “Boring theoretical knowledge combined with psychological pressure will further affect their academic performance.” It sounds like the authors are stating that theoretical knowledge is boring. This can be offensive to academics and teachers who over theoretical concepts. It may be better to restate that students may perceive subjects as boring or change the word all together.

Response: Thank you for the suggestion, we have removed this sentence.

Methods

  1. The sports participation frequency variable operationalization was well explained; however, the authors state that the validity can be reviewed “elsewhere.” I would suggest that the authors provide some report of the prior validity given this is the primary independent variable of interest in all of the models.

Response: Thank you for the suggestion, the sports participation is based on the item of Youth Risk Behavior Survey (YRBS); this item has been fully applied in different countries and has sufficient reliability and validity. More information regarding YRBS procedures can be found by accessing the Centers for Disease Control and Prevention website.

  1. Can the authors clarify the academic performance variable. This is only the grade on a single exam or is this their grade in the class. Also please clarify if this is a self-reported grade. If this is self-report please indicate in the limitations and potential bias that can be inherent in self-reported exam grades.

Response: Thank you for the suggestion. We have added the limitation in the study, please see the line 436-439.

  1. Regarding the modeling approach what are the authors reporting odds ratios for the linear models? Would it be simpler to report betas for the linear change? Please explain.

Response: Thank you for the suggestion, the academic achievement of outcome is a grade variable, and it is more appropriate to explain it with OR.

  1. Given that the data was collected in schools and classrooms did the authors consider multilevel models? There are two elements of concern: 1) the models may be identifying class differences or school differences that are unaccounted for. The clustering at the school or classroom level may be biasing the model effects and 2) given the large sample size are the models overpowered? Sometimes overpower in a model can drive significance. Was a power analysis run? Partitioning the groups in a multilevel model can also help here too. I am not saying this is a requirement just needs to be explained and maybe considered in the limitations if not addressed. What was the ICCs across schools or classrooms? Where they at an acceptable level.

Response: Thank you very much for this comment. As a matter of fact, we first used multilevel models in our study. However, the ICC coefficient was extremely close to 0.00, indicating that multilevel models were not necessary. That is the reason why we did not perform a multilevel modeling analysis.

Results

  1. Given that some of the Cis are not overlapping could some of these results suggest a dose-response relationship? Maybe this can be considered in the discussion. If so how much is a good amount?

Response: Thank you for the suggestion, please see the line 354-362.

Discussion

  1. I recommend that the authors take care in conclusive statements regarding the results given the cross-sectional nature of the study design. Example: “In sum, it is found in the present study that sports participation can effectively improve the academic performance of children.”

Response: Thank you for the suggestion, we have re-edited this sentence. Please see the line 303.

  1. Could the authors discuss some about the type of activity? I know that this was not the focus of the current studies results but, discussion about the types of activity or specifics about recommended activity durations per week/day.

Response: Thank you for the suggestion, we have added the type of activity, please see the line 457-464.

  1. Consider adding some to the limitations based on prior comments from above.

Response: Thank you for the suggestion. We have added the limitation in the study.

Round 2

Reviewer 2 Report

The authors have addressed all of my concerns. 

Author Response

Thank you very much for your comments!